# Non-thermal emission in gap-mode plasmon photoluminescence

Robert Lemasters [1] ✉, Manoj Manjare[1], Ryan Freeman[1], Feng Wang[1], Luka Guy Pierce[1], Gordon Hua[1], Sergei Urazhdin [1] & Hayk Harutyunyan [1] ✉

Photoluminescence from spatially inhomogeneous plasmonic nanostructures exhibits fascinating wavelength-dependent nonlinear behaviors due to the intraband recombination of hot electrons excited into the conduction band of the metal. The properties of the excited carrier distribution and the role of localized plasmonic modes are subjects of debate. In this work, we use plasmonic gap-mode resonators with precise nanometer-scale confinement to show that the nonlinear photoluminescence behavior can become dominated by non-thermal contributions produced by the excited carrier population that strongly deviates from the Fermi-Dirac distribution due to the confinement-induced large-momentum free carrier absorption beyond the dipole approximation. These findings open new pathways for controllable light conversion using nonequilibrium electron states at the nanoscale.

Noble metal nanostructures exhibit remarkable broadband photo-luminescence (PL) under near-infrared (NIR) pulsed excitation, spanning both up- and down-converted spectral regions[1–3]. This emission is attributed to the generation of energetic electrons in the conduction band, leading to a radiative recombination process. These electrons are commonly described as hot, under the assumption that they form a quasi-equilibrium Fermi-Dirac distribution with a large effective temperature. Hot-electron injection and dynamics have recently attracted considerable interest due to their potential applications in light harvesting, light-matter interactions, and optically driven catalysis[4,5]. Additionally, hot carriers exhibit ultrafast dynamics that can be used in all-optical switching[6–8]. These applications of the broadband PL require understanding of the underlying mechanisms of light-driven nonequilibrium electron dynamics.

For the incident photon energy above the interband transition threshold, down-converted PL results from the recombination of $sp$ conduction-band electrons with $d$-band holes in bulk Au[9]. When the incident photon energy is below the interband energy, such as for the NIR excitation, direct single-photon interband absorption is not allowed. In this regime, PL is facilitated by multiphoton absorption (Fig. 1, right panel), which can be characterized by the dimensionless nonlinear power-law exponent (PLE) $p$ relating emission to the excitation intensity. In early studies on rough Au films, up-converted emission was shown to scale quadratically with the excitation power ($p = 2$), which was attributed to two-photon interband PL (2PPL). Meanwhile, the down-converted PL scaled linearly ($p = 1$)[1]. The underlying process was speculated to be similar to Landau damping in plasmon-mediated single-photon intraband absorption, wherein large field gradients of localized modes enable efficient transitions characterized by large momentum transfer typically forbidden in the dipole approximation (Fig. 1)[10–12].

Subsequent studies of similarly rough metal films showed that $p$ can be non-integer, indicating that the mechanisms of non-linear PL (NPL) may be more complex[3]. The power-law exponent was found to increase approximately linearly with the emitted photon energy $\epsilon = \hbar\omega$, $p(\epsilon) \propto \epsilon$. These findings were explained as follows: Plasmon absorption initially leads to a non-thermal excited electron population that rapidly equilibrates into the Fermi-Dirac distribution with an effective temperature reaching several thousand Kelvin (Fig. 1, left panel). Subsequent recombination results in broadband emission with frequency-dependent PLE, $p(\epsilon) = \epsilon/ak_{B}T_{\text{eff}}$, characterized by a single thermodynamic state variable, the effective temperature $T_{\text{eff}}$ of the electron gas. Here, $k_{B}$ is the Boltzmann constant and $a$ is heat capacity.

An isolated single nanostructure avoids ensemble averaging that can obscure the intrinsic mechanisms of NPL, which has motivated the studies of broadband emission from single nanoparticles, dimers, and few-particle aggregates[13–22]. These studies showed that the hot-electron

[1]Department of Physics, Emory University, Atlanta, GA 30322, USA. ✉e-mail: robertdlemasters@gmail.com; hayk.harutyunyan@emory.edu

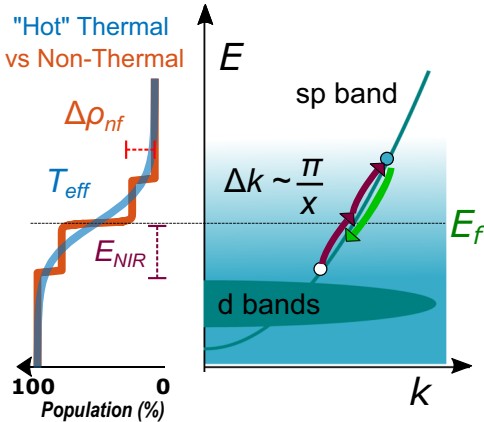

**Fig. 1 | Hot carrier generation in noble metals.** Near-infrared (NIR) intraband multiphoton absorption (upward arrows) and emission (downward arrow) mechanism induced by the wavevector of the localized plasmon mode. The resultant "hot" (thermal) vs non-thermal electron distributions with the corresponding state variables.

intraband model, while accounting for the PL emission from rough films, may not provide a good approximation for isolated nanostructures. Furthermore, these studies pointed to a plethora of possible processes dependent on the specific experimental conditions, such as nanoparticle geometry and the choice of excitation source[3,23]. Additional channels such as multiphoton absorption, hybrid interband-assisted transitions[2,20,24,25], and Raman scattering[26–29] were invoked to explain the observed spectral characteristics of PL emission and their power scaling, suggesting that the underlying mechanisms may not be universal. Broadband PL was recently observed even under CW excitation, putting into question the roles of field gradients and of the overall field enhancement[19,20,30].

A common theme among these experiments is the requirement for large momentum transfer in intraband transitions facilitated by field confinement. Transitions characterized by large momentum transfer were recently suggested to play a central role in the coupling of propagating plasmons to quantum emitters and quantum wells[31,32]. Transient optical reflectivity changes of metal films were explained by Landau damping induced by the large-wavevector components of optical fields[33,34]. Large-momentum transitions have also been explored as a possible mechanism of resonant absorption in indirect-gap semiconductors such as Si[35,36], which was supported by indirect experimental evidence[37]. A similar absorption mechanism was also proposed for extremely small, nanometer-size particles[38–42].

These studies pointed to the importance of large momentum transfer for optical transitions, but did not provide a direct experimental confirmation for this mechanism. In particular, geometric limitations of nanoparticle systems have prevented systematic probing of the effects of field confinement on NPL. For plasmonic structures, the characteristic wavevector of the optical field is $k \backsim \pi/x$, where $x$ is the length scale associated with the mode[1]. On the other hand, the $k$-vector mismatch for intraband transition in Au determined by the Fermi velocity is of the order of $(5\,\text{nm})^{-1}$ [43]. Thus, the characteristic dimensions that control plasmonic near fields must be in a few-nanometer range to generate significant field components with $k$-vectors required for intraband transitions[11,44]. The characteristic dimensions of plasmonic modes in colloidal particles, and especially in dimers, can reach this limit. However, the geometric uncertainties inherent to this nanostructure can result in significant variations among individual systems, which similarly to rough films masks the influence of sharp, high-$k$ field gradients versus field enhancement on broadband PL emission.

In this work, we study NPL in metal-dielectric-metal (MDM) nanogaps. Precisely controlled field confinement in the studied nanogaps enables the formation of a non-Fermi transient electron distribution driven by the plasmonic field, due to the breakdown of the dipole approximation. Our approach enables efficient control of the relative strength of thermal and non-thermal carrier contributions to NPL by engineering large-momentum electronic transitions in nanostructures.

## Results

### Experimental overview

The studied MDM structures consist of a 40 μm by 40 μm array of Au nanowires with a periodicity of 350 nm, fabricated on top of an Au film and separated from it by a few nanometer-thick dielectric spacer[45]. This geometry supports gap plasmon modes, allowing one to control the spatial confinement of plasmonic fields by varying the dielectric spacer thickness (Fig. 2a). This design also allows us to correlate the PL signal with the plasmonic effects, by taking advantage of the fact that only incident light polarized perpendicular to the nanowire excites the plasmon mode (Supplementary Fig. 2).

The intensity of PL follows the plasmonic scattering spectrum. In particular, the PL intensity is maximized when the laser frequency is on-resonance with the plasmon mode[15]. In our MDM geometry, the plasmon resonance frequency depends on both the wire width, $W^{40}$, and the spacer thickness, $d_{\text{SiO}_2}$, with the latter dependence arising due to the proximity interactions with image charges in the metal underlayer[45] (Fig. 2b). The nanowires are gently tapered (Fig. 2a), with the wire width varying by $\Delta W = 30$ nm between the two ends of the wires, resulting in a gradual variation of the resonance frequency along the wire.

The small taper allows fine tuning of plasmon resonance over a broad wavelength range $\Delta\lambda \backsim 300$ nm ($\Delta\epsilon = 0.53$ eV), so that the gap-mode resonance is in the desired range of PL wavelengths 750–1050 nm detected in our measurements (Fig. 2c). To precisely control the field confinement in the dielectric gap, we fabricate ultra-smooth Au surfaces using cryogenic sputtering deposition, which minimizes surface roughness[46], followed by atomic layer deposition of SiO$_2$ dielectric spacer layer (see Methods for details on sample fabrication). Representative PL emission spectra from these structures shown in Fig. 2d clearly exhibit nonlinear wavelength-dependent scaling of PL. In particular, doubling the excitation intensity results in approximately doubled down-converted light intensity ($p \approx 1$), whereas the up-converted emission intensity scales super-linearly and approximately triples.

To study the effect of light confinement on the emission spectra, the samples were fabricated with different SiO$_2$ spacer thicknesses $d_{\text{SiO}_2}$ of 1, 2, 3, and 15 nm. For each thickness, the width of the wires was adjusted to obtain the same spectral position of the plasmon peak in all four samples (Fig. 3a). For the NPL measurements, the MDM nanostructures are excited with a 785 nm Ti:Sapph pulsed laser, with the position along the wire adjusted so that it is on-resonance with the gap-mode plasmon (see Supplementary Fig. 1 and Methods for the details of experimental setup). The laser has a 150 fs pulse duration and 80 MHz repetition rate. A maximum peak power of $\backsim 0.1$ GW/cm$^2$ was used in these experiments. To determine the PLE, $p(\epsilon)$, the emission spectra, $\phi(\epsilon)$, are recorded for several values of laser power. $p(\epsilon)$ is then calculated by fitting the dependence of the spectrally-resolved intensity $\phi(\epsilon)$ on the laser power, $I$, with $\phi(\epsilon) \propto I^{p(\epsilon)}$.

### Gap-dependence of PL

The values $p(\epsilon)$ of PLE depend both on the emission frequency and the spacer thickness $d_{\text{SiO}_2}$, decreasing with decreasing $d_{\text{SiO}_2}$, Fig. 3b. Qualitatively, this is consistent with stronger field confinement, which increases the efficiency of single-photon excitation at small incident power by providing momentum matching between the field and

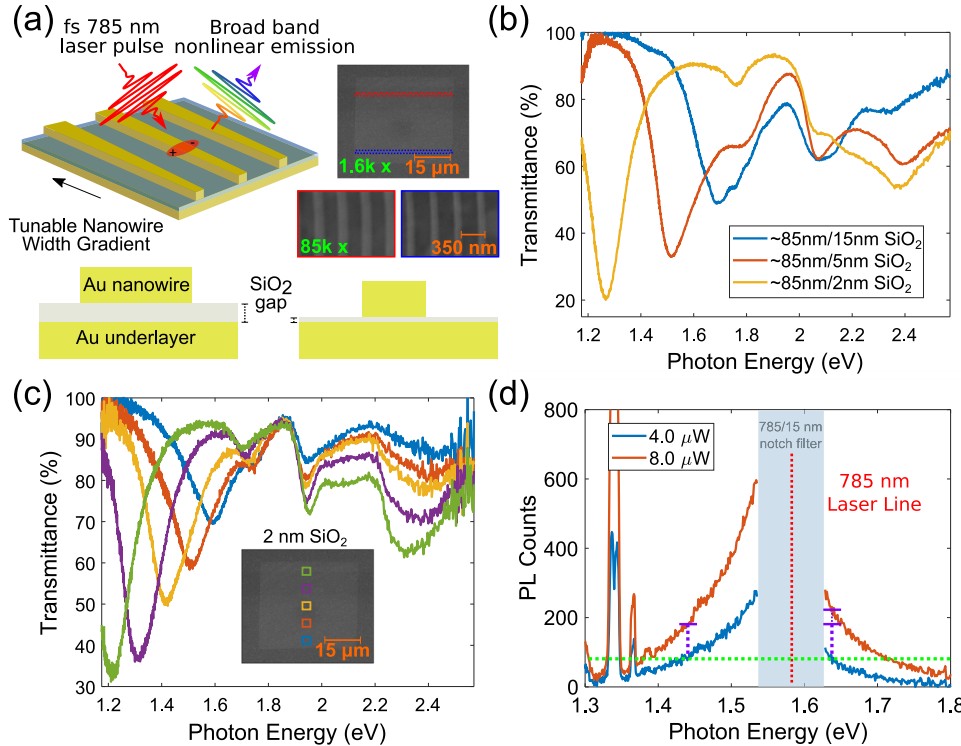

**Fig. 2 | Optical properties of metal-dielectric-metal (MDM) cavities. a** Schematic of the tapered nanowire array in the MDM geometry and representative SEM images. Cross-section of MDM structures with different dielectric layer thicknesses. **b** Redshifting of plasmonic resonance with decreasing spacer thickness, for an ~85-nm-wide nanowire. **c** Broad tunability of gap-mode resonance by varying the width of the nanowires. **d** Typical NPL emission spectrum.

electron excitation, resulting in a smaller PLE. In the hot-electron picture, larger characteristic plasmonic wavevectors enable the excitation of higher-energy electrons characterized by a larger effective temperature $T_{\text{eff}}$, resulting in a smaller PLE[3].

Intriguingly, the PLE exponents exhibit a pronounced dependence on the emitted photon energy and the spacer thickness. The dependence for the down-converted spectral range is substantially different from that for the up-converted range. In the down-converted range, PLE approximately linearly increases with increasing photon energy, with the slope that increases with increasing spacer thickness. A rough Au film control sample exhibits the largest slope, signifying the weakest mode confinement. This result is in agreement with previous studies[3]. In the up-converted range, PLE increases with emitted photon energy, following the trend that can be smoothly extrapolated from the down-converted range, but saturating at large energies.

Several previously proposed mechanisms, including 2PPL, hot-Raman scattering, and thermal intraband PL, may, in principle, contribute to this dependence (see Supplementary Section "Modeling and Simulations" for their detailed discussion). However, these mechanisms alone are insufficient to explain the observed behaviors of nonlinear emission, as illustrated in Fig. 3c for the representative spacer thickness $d_{\text{SiO}_2}$ = 3 nm. The only reasonable fit to the experimental dependence is provided by the intraband PL with non-Fermi electronic distribution (purple line). Furthermore, when more than one contributing process is considered (Fig. 3d), an even better fit is obtained by including both thermal and non-Fermi PL, whereas hot-Raman and 2PPL provide worse fits when combined with the non-Fermi emission. Thus, the observed emission can be well-approximated by

$$\phi(\epsilon) \propto \alpha\phi_{\text{th}}(\epsilon) + (1 - \alpha)\phi_{\text{nf}}(\epsilon), \tag{1}$$

where $\phi_{\text{th}}$ is the thermal emission whose relative weight is described by $0 \le \alpha \le 1$, and $\phi_{\text{nf}}$ is the non-thermal (not described by the Fermi distribution) emission with the relative weight $1 - \alpha$.

## Non-thermal (non-Fermi) PLE

It is usually assumed that the non-Fermi distribution−transient regime of photoexcitation that cannot be described by an effective temperature−does not provide a substantial contribution to the PL yield or the spectral dependence of PLE, for two main reasons. First, the initial non-Fermi distribution is extremely short-lived compared to the longer ps-scale relaxation time of the hot electrons, and thus the latter regime is expected to dominate PL[3]. Second, the transient carrier distribution is flat, and does not produce power-dependent spectral features[16].

The first argument is based on the assumption that the recombination times are much longer than the carrier thermalization time. This is likely the case for typical plasmonic nanostructures where thermalization occurs on the timescale of tens of fs[47−50]. However, for extreme light localization in the nanogaps, the oscillator strength of the intraband transitions can be dramatically enhanced due to the breakdown of the dipole approximation, leading to fast recombination with rates comparable to the thermalization rate.

The second argument overlooks the fact that the initial non-Fermi distribution forms steps at energies shifted above and below the Fermi level by the energy of the incident photon (Fig. 1). Furthermore, quantum analyses, including the effects of bandstructure and the geometry of the system, predict far more complicated non-thermal carrier distributions than the step-like function in Fig. 1d[12,50]. To the best of our knowledge, precise numerical modeling of the PLE lineshape for such non-Fermi distributions has not yet been attempted, although several studies have investigated the spectral character of emission with a non-Fermi term[22,30].

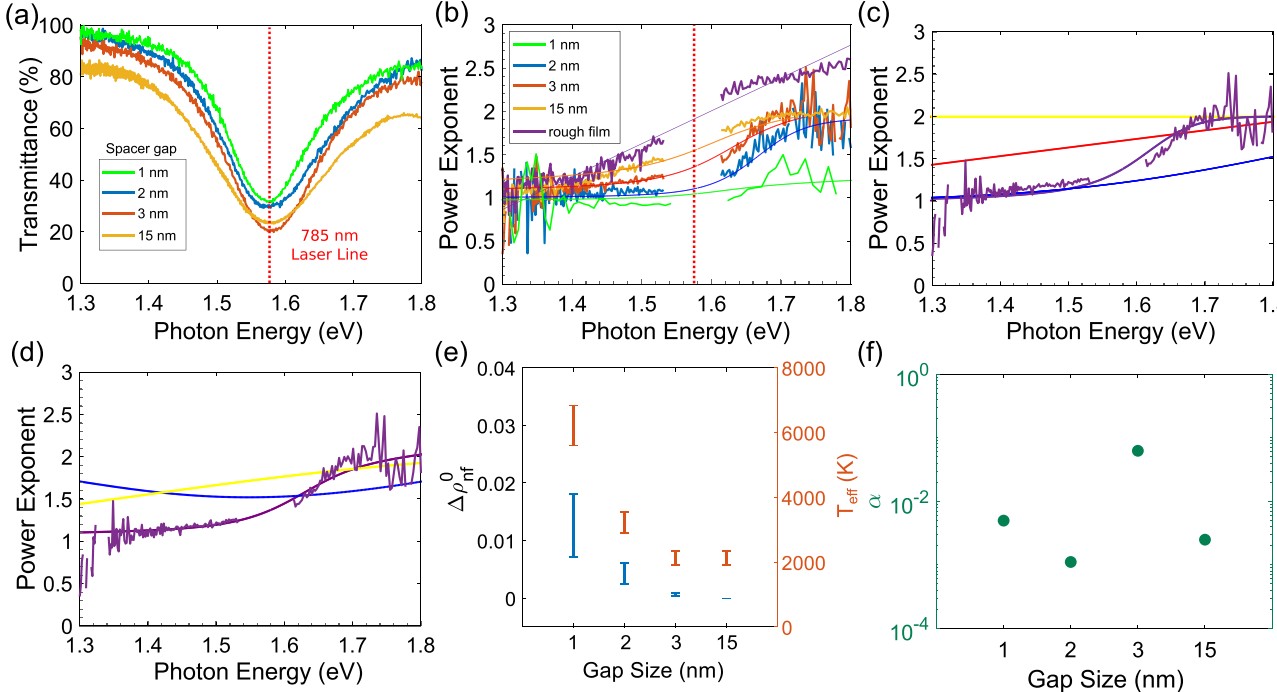

**Fig. 3 | Gap-dependence of PL. a** Tuning and matching of gap-mode plasmon resonances among various SiO₂ spacer gap thicknesses. The resonant energies are equal to the laser energy for maximum absorption. **b** Photoluminescence (PL) power-law exponents (PLE) vs emitted photon energy, for the labeled values of $d_{SiO_2}$. Smooth curves are fits, as described in the text. **c, d** PLE dependence on emitted photon energy for $d_{SiO_2} = 3$ nm. Smooth curves in (**c**) are dependences expected for different nonlinear PL (NPL) emission mechanisms: "hot" Raman emission (blue), intraband "hot" electron emission (red), two-photon interband PL

(2PPL) emission (yellow), and non-thermal emission (purple). Smooth curves in (**d**) are dependences expected for hybrid "hot" electron NPL emission mechanisms: "hot" electron/"hot" Raman scattering (blue), "hot" electron/2PPL (yellow), "hot" electron/non-thermal (purple). **e** Fitting parameters obtained from hybrid "hot" electron/non-thermal fitting of gap PLE in (**b**). **f** Extracted ratios of the relative contributions of thermal to non-thermal (thermal:non-thermal) electronic distributions from the emitted light as a function of gap size.

Here, we derive the expression for the PLE lineshape for the non-Fermi distribution given by[51]

$$f_{nf}(\epsilon) = f_0(\epsilon) + \Delta\rho_{nf}(\epsilon), \quad (2)$$

$$\Delta\rho_{nf}(\epsilon) = \Delta\rho_{nf}^0 \left\{ f_0(\epsilon - \epsilon_p)[1 - f_0(\epsilon)] - f_0(\epsilon)[1 - f_0(\epsilon + \epsilon_p)] \right\}, \quad (3)$$

where $f_0$ is the equilibrium distribution prior to absorption ($T = T_{amb}$), $\Delta\rho_{nf}^0$ is the amplitude of the population change determined by the intensity of the light source, and $\epsilon_p$ is the excitation laser photon energy. Energy is counted from the Fermi level, $\epsilon_F = 0$. To estimate the non-Fermi PL, we follow the approach of ref. 3, where the restrictions on the wavevector are neglected, so that the Fermi's golden rule for the transition probability is reduced to a simple overlap integral of the electron and hole distributions. We extend this approach to the non-Fermi distribution $f_{nf}$, yielding the emission spectrum

$$\phi_{nf}(\epsilon) \propto \int_{-\infty}^{\infty} f_{nf,e}(\epsilon', T_0) \rho(\epsilon) f_{nf,h}(\epsilon' - \epsilon, T_0) \, d\epsilon', \quad (4)$$

where $f_{nf,e}$ is the electronic non-Fermi distribution, $f_{nf,h} = 1 - f_{nf,e}$ is the hole non-Fermi distribution, $\rho$ is the photonic density of states approximated by the plasmonic scattering spectrum, and $T_0$ is the ambient temperature if the thermalization is neglected for sufficiently short time scales. We solve Eq. (4) numerically to obtain

$$\phi_{nf}(\epsilon) \propto A(\epsilon) + B(\epsilon)\Delta\rho_{nf}^0 + C(\epsilon)(\Delta\rho_{nf}^0)^2, \quad (5)$$

where, $A$, $B$, & $C$ are numerical weighting functions. Eq. (5) includes linear and quadratic terms in the excitation power, since $\Delta\rho_{nf}^0$ scales linearly with power[51]. This approach provides a very good agreement with the observed PLEs (Fig. 3b), capturing the distinct behaviors observed in the up- and down-converted spectral regions, including the asymptotic values $p \approx 2$ at large energies and $p \approx 1$ at small energies. The agreement is further improved by including the hot-electron contribution in addition to the non-thermal terms of Eq. (5) (Fig. 3d). Fitting of the gap-dependent PL (Fig. 3e) shows that both $\Delta\rho_{nf}^0$ and $T_{eff}$ increase with decreasing spacer thicknesses, consistent with the expected enhancement of Landau damping with increasing confinement. However, the emission is completely dominated by the non-thermal contribution, as evidenced by very small values of $\alpha$ for all spacer thicknesses (Fig. 3f).

## Detuning-dependent PL

We confirm the proposed interpretation of the observed PLE lineshape in terms of a non-thermal distribution produced by the large momentum transfer by analyzing its dependence on several parameters tunable in our experimental approach. Two main parameters characterize the fields in a nanogap: field enhancement and field confinement. In the samples discussed above, the electric field in the gap is expected to substantially increase with decreasing gap width. Finite-difference time-domain method (FDTD) simulations of the electric field in these geometries demonstrate that the relative field enhancement, compared to free-space light, approaches ≈20 for $d_{SiO_2} = 15$ nm, further increasing to ≈70 for $d_{SiO_2} = 2$ nm (see Supplementary Section "FDTD Simulations" and Supplementary Fig. 5 for detailed results). Thus, the gap thickness affects both the field enhancement and field confinement. To independently control the field enhancement without affecting the confinement, we detune the

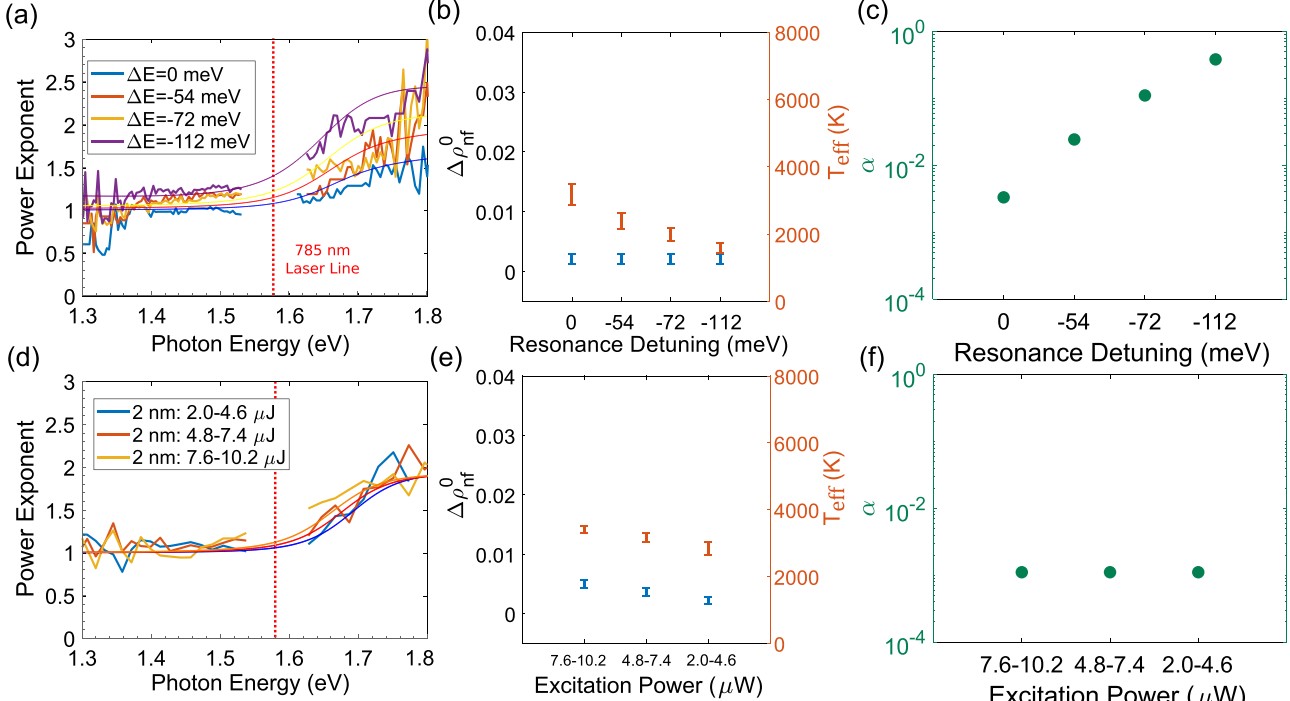

**Fig. 4 | Detuning- and fluence-dependent photoluminescence (PL). a** Power-law exponent (PLE) as a function of emitted photon energy extracted from the non-linear PL (NPL) at different resonance detuning for $d_{SiO_2} = 2$ nm, as labeled. **b** Fitting parameters used in PLEs in (**a**). **c** Extracted ratios of the relative contributions of thermal to non-thermal (thermal:non-thermal) electronic distributions from the emitted light as a function of detuning. **d** PLE as a function of emitted photon energy extracted from the NPL, at different laser excitation power, as labeled. **e** Fitting parameters used in PLEs in (**d**). **f** Extracted ratios of the relative contributions of thermal to non-thermal (thermal:non-thermal) electronic distributions from the emitted light as a function of fluence.

plasmonic resonance away from the excitation photon energy while keeping the spacer thickness constant. This allows us to investigate whether the observed dependence of the PLE lineshape on the gap width can be explained by the effects of field enhancement rather than the proposed field confinement[18,20].

To investigate the effects of off-resonance excitation, we detune the plasmon mode from the laser energy by shifting the excitation beam along the tapered wire. The characteristic wavevector determined by the gap remains constant, and there is still substantial spectral overlap between the plasmon mode and the laser (see representative spectra in Supplementary Fig. 5h), but the plasmonic field enhancement is reduced. FDTD simulations confirm that field enhancement is reduced with increasing detuning. For instance, for $d_{SiO_2} = 3$ nm, the field enhancement drops by a factor of $\approx 4$ at detuning $\Delta E = -112$ meV as compared to the on-resonance excitation (see Supplementary Fig. 5e–h).

Figure 4a shows the dependences of PLE on the emitted photon energy for several values of detuning, for $d_{SiO_2} = 2$ nm. These data show that the overall amplitude of $p(\omega)$ increases with increased detuning consistent with the dependence of the overall efficiency on the field enhancement, with lower $p$ corresponding to higher efficiency. Both the non-thermal population $\Delta\rho_{nf}^0$ and the effective temperature of the thermal contribution $T_{eff}$ show only a modest dependence on detuning (Fig. 4b). More notably, the relative weight $\alpha$ of the thermal contribution dramatically increases with detuning (Fig. 4c). This is consistent with the importance of large field gradients associated with the highly confined cavity modes for the generation of a substantial non-thermal carrier population. Non-thermal contribution is most pronounced at resonance, when the field is strongly confined, whereas the thermal contribution does not require resonant mode excitation. As resonance detuning increases, the relative contribution of non-thermal contribution decreases. As a consequence, $\alpha$ increases and approaches unity at large detuning.

## Fluence-dependent PL

An additional test for our interpretation is provided by the dependence of PLE $p(\varepsilon)$ on the excitation fluence. According to the analysis presented above, $\Delta\rho_{nf}^0$ should scale linearly with fluence, while $T_{eff}$ should scale as an exponent of $1/a^3$. To test these scaling relationships, we have repeated the PLE measurements for several distinct characteristic fluence values by breaking down the full excitation power range into narrow non-overlapping segments.

The measured dependences are consistent with the expected scaling relations (representative emission spectra for $d_{SiO_2} = 2$ nm are shown in Supplementary Fig. 3c) The PLE lineshapes vary only slightly with fluence (Fig. 4d), and the values of $\Delta\rho_{nf}^0$ and $T_{eff}$ obtained from their fitting (Fig. 4e) scale as expected from our model. The values of $\alpha$ (Fig. 4f) remain small, indicating that the process is dominated by the non-thermal population for the thin spacer. This excellent agreement of the data with our model supports the validity of the proposed picture of non-Fermi electron contribution to NPL.

## Discussion

We have provided the first, to our knowledge, direct demonstration of a significant contribution of non-Fermi electronic distribution to NPL emission from spatially confined systems comprising two metallic electrodes separated by nanometer-scale gap confining the optical field. Signatures of non-thermal distribution are evident for gap sizes of up to 15 nm, where the thermal contribution starts to dominate. The most pronounced effect is manifested by the dependence of emission on the excitation power. In the regime dominated by non-thermal distribution, the corresponding exponent rapidly varies from $p = 1$ (linear relation) at emitted photon frequencies below the excitation frequency, to $p = 2$ (quadratic relation) above the excitation frequency. Increasing electron thermalization results in an increasingly gradual variation of $p$ with emitted photon frequency, well-approximated by a linear dependence for sufficiently small emitted photon frequency

energy range, on the scale determined by the temperature of thermalized "hot" electrons. By comparing experimental PLE lineshapes to models such as multiphoton and Raman scattering, we were able to rule out these mechanisms as the possible origin of NPL in noble metals.

Non-thermal effects are most pronounced for small gap sizes, consistent with the expected enhancement of Landau damping. In contrast, non-thermal contribution becomes negligible for standalone rough metal films, consistent with the weak localization effects due to film roughness. Efficient excitation of non-Fermi electronic distribution that can dominate radiative recombination in strongly confined geometries is promising for applications in photovoltaics, hot carrier-induced catalysis, and spasers.

## Methods
### Gradient nanowire arrays
Glass substrates are sonicated in a solution of deionized (DI) water and detergent for 20 min at 50 °C, rinsed with DI water, and followed with a 20 min acid wash using a 3:1 piranha acid solution of $H_2SO_4$ and $H_2O_2$. Substrates are then thoroughly rinsed in DI water, dried with nitrogen, and finally stored in isopropyl alcohol. The Au-spacer sample is fabricated by first cryogenically sputtering 30 nm of Au onto a glass substrate. This was followed by sequential monolayer deposition by plasma atomic layer deposition (ALD) to produce the $SiO_2$ dielectric spacer layer of varying thickness.

The nanowire arrays are fabricated by using standard positive resist electron-beam lithography (EBL) using a spin-coated MMA and PMMA-950 bilayer for the bottom and top resist layers, respectively.

Following EBL, the sample is developed in a 1:1 solution of MIBK:IPA for 15 sec at 0 °C. Differential contrast microscopy is used to monitor and confirm the development of the polymer film via color contrast, as occasional inhomogeneities of the PMMA/MMA thickness necessitate further development in MIBK:IPA. Once pattern development is confirmed, another 30 nm layer of Au is deposited by thermal evaporation, then the sample is put under acetone to remove any excess polymer. The excess metal film is rinsed off and the finished sample is transferred to isopropyl alcohol and then blown dry with $N_2$ gas. The microarrays are then examined using a Zeiss scanning electron microscope (SEM) to ensure the quality and geometric properties of the nanostructures.

### Cryogenic sputtering
Surface roughness on the Au underlayer can add additional unwanted PL signal from the sample and contaminate the optical signal from the gap-mode plasmons. Coinage metals deposited on oxide substrates have poor adhesion and typically will result in roughened metal surfaces for thin films. To produce films which are smooth as possible, an in-house cryogenic sputtering system was employed to produce ultrasmooth Au underlayers[46]. The glass substrate is fixed to a cold finger, which puts the substrate into a thermal bath with a liquid nitrogen reservoir, which is maintained throughout the entire sputtering procedure. The temperature of the substrate is confirmed to be ∽−195 °C via a thermocouple monitor prior to deposition. Deposition is then performed at ∽$10^{-7}$ Torr with a rate of 5 Å/s for 60 sec resulting in a 30-nm film. Atomic force microscopy (AFM) is then performed for metrology of the Au film surface roughness, which is found to be reproducibly below ∼3 Å.

### Atomic layer deposition
Atomic layer deposition (ALD) is performed with a Cambridge FIJI Plasma ALD system to produce the nanometric dielectric spacer layers. Fabrication of these layers are performed in a class 100 cleanroom to prevent contamination during the process. ALD is more favorable than other deposition techniques, such as thermal evaporation and sputtering, as it produces pinhole-free films with ultra-high aspect ratio features[52]. The $SiO_2$ ALD process is plasma assisted and deposited at rate of ∽0.6 Å/cycle with a chuck temperature of 150 °C. The thicknesses are confirmed by both ellipsometry measurements and an AFM "scratch" test. $SiO_2$ ALD is done using tris(dimethylamino)silane and $O_2$ precursors.

### Optical measurements
The experimental setup is shown in Supplementary Fig. 1. For transmittance measurements, white light is provided by a halogen lamp. An iris is placed just after the lamp to limit the angular distribution of light rays incident on the sample. This is done so as not to excite any higher-order grating modes which could potentially overlap with the fundamental gap-mode spectral region. The MDM nanowire array is placed on a piezo stage, as fine tuning of the position along the gradient gratings is crucial for the selection of the gap-mode center resonance.

The scattered light is collected by a 60x NA = 1.42 oil objective for maximal light collection. The MDM structure is immersed in the oil of the objective, facing the collection path. This serves to significantly reduce long-term oxidation effects on the dielectric spacer layer, which can result in unwanted surface roughness, and utilizes the Au under layer to act as a mirror for the light emission, and so enhances the signal yield. This approach also serves to eliminate undesirable Fano lineshapes produced from interference between the scattered and incident light fields due to an induced phase mismatch upon passing through the surface[53].

The signal is then passed through an expanding lens before passing through a polarizer, which is aligned to the polarization-dependent resonance of the nanowire array. Finally, the light is sent through a collimating lens before it is passed through two perpendicular 150 μm slits (one of which is internal to the spectrometer and adjustable), which act as a pinhole to confocally isolate the collected signal from the MDM structure, ensuring the signal is isolated to only the spatially and spectrally tuned nanowires. The spectrometer is an Andor iDus with a 1024 by 128 pixel CCD chip array cooled to −70 °C. The transmittance is then calculated by $T = (\phi_{sig} - \phi_{dark})/(\phi_{ref} - \phi_{dark})$, where $\phi_{sig}$ is the spectrum from the MDM region of interest, $\phi_{ref}$ is the reference halogen lamp spectrum from an empty location on the Au film away from the MDM structures and $\phi_{dark}$ is the dark spectrum of the spectrometer

For NPL measurements, a Ti:Sapph pump provides a 785 nm wavelength and 150 fs pulsed laser output at an 80 MHz repetition rate. The pulse is sent through an optical parametric amplifier (OPA) system where it is simply passed through and unaltered spectrally, but is reduced to 20% of it's initial intensity with an output from the laser window of ≈20 mW. The output intensity is again lowered by a fixed neutral density (ND) filter by 2 orders of magnitude before being passed through a 785 ± 15 nm single-band pass filter to eliminate any sidebands of the laser pulse as well as any residual luminescence from the Ti:sapph laser. The pulse is then passed through a variable ND filter for fine tuning of the power intensity for performing power-dependent emission measurements. The power fluences used for NPL experiments on gap modes were over a range of ≈2−16 μW. Given the pulse length, repetition rate, and spot size, this corresponds to peak powers of ≈ 0.01−0.05 GW · cm$^{-2}$ at the excitation spot. A 50-50 beam splitter sends the pulse to the objective where it is focused on the MDM structure where it has been previously positioned such that the plasmonic resonance is at the desired value at the laser spot location.

The continuum NPL emission is collected through the same objective where it is then passed through the 50-50 beam splitter and sent to the emission filters. The emission is sent through a 785 nm notch filter which rejects wavelengths of 785 ± 15 nm to remove any residual signal from the laser, followed by a polarizer, which is aligned to the polarization-dependent resonance of the nanowire array. Finally, the emission passes through a collimating lens, and before entering the spectrometer, the external slit is removed, and the spectrometer slit is set to 200 μm to increase signal count while still maintaining decent

spectral resolution. The integration time of the signals is 2 min for all spectra, with the exception of low fluence measurements, which used 10 min times due to the small emission signal. All PL measurements are performed in black-out conditions. The PL signal, $\phi$, is calculated by $\phi = \phi_{sig} - \phi_{dark}$, where $\phi_{sig}$ is the emission spectrum from the MDM region of interest and $\phi_{dark}$ is the signal with the laser blocked from the entire optical path. The transmittance of the entire emission collection path, including the beam splitter and polarizer was also recorded and used to normalize the PL signals.

## Statistics and reproducibility

The laser pulse is polarized and the polarization-dependent MDM structures are aligned accordingly (Supplementary Fig. 2a). This allows for the correlation of the resulting PL with the gap-mode resonance. To confirm the integrity of the resonators after exposure to high peak power laser pulses, the transmittance of the plasmon resonance is taken both before and after pulsed laser excitation to ensure no spectral shifting due to photothermal melting/reshaping of the nanowires occurred. The large field enhancement in the gap combined with the relatively large peak power of the laser can indeed cause this and is observed (Supplementary Fig. 2b, c). Subsequent measurements are limited to be lower than this damage threshold range. If the resonance is seen to have shifted or altered after exposure to laser pulse excitation, the data sets are excluded due to concern of laser-induced restructuring.

Multiple MDM samples were fabricated and displayed analogous nonlinear behavior for a given gap geometry. The integration times were also varied and included 1, 2, 5, 10, and 60 min, in addition to the 2 min exposure times used for the majority of the data presented in this study. The longer times resulted in better signal to noise of the emission spectra and did not display any dependence to exposure times.

## Power-law exponent calculations

Raw emission spectrum data, $\phi(\epsilon)$, was processed and analyzed in MATLAB. Using the nonlinear scaling relationship with laser intensity, $I$: $\phi(\epsilon) \propto I^{p(\epsilon)}$, the PLE lineshapes, $p(\epsilon)$, were extracted via a linear fit to logarithmic plots of intensity-dependent emission at a given photon energy, $\epsilon$, by $p(\epsilon) = d \log \phi(\epsilon)/d \log I$.

Numerical calculations for power-law exponent lineshapes were performed in Mathematica using the parameters and models described in the main text. A table of the fitting parameters and their corresponding ranges for numerical fitting to each experimental condition is given in Supplementary Table 1.

## Reporting summary

Further information on research design is available in the Nature Portfolio Reporting Summary linked to this article.

## Data availability

The data that support the findings of this study is available in the Figshare repository with the digital object identifier https://doi.org/10.6084/m9.figshare.25483102.

## Code availability

The MATLAB script for power-law fitting to data is available in the Figshare repository with the digital object identifier https://doi.org/10.6084/m9.figshare.25483102. The Mathematica code used to generate theoretical fits are available from the corresponding authors.

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

## Acknowledgements
This work was supported by the US Department of Energy award No DE-SC0020101 and by the NSF award ECCS-2005786.

## Author contributions
The concept and experimental design were developed by R.L. and H.H. Data acquisition was performed by R.L. The FDTD simulation analysis was conducted by R.L. and F.W. The cryogenic sputtering technique was developed by R.L. and H.H., with the sputtering chamber constructed by M.M., R.F., and S.U. Electron-beam lithography was carried out by R.L., with significant consultation from R.F., H.H., and S.U., and optimization by R.L. and G.H. Sample fabrication was undertaken by R.L. and M.M. Substrate and film metrology were conducted by R.L. and L.G.P. Data analysis was carried out by R.L. The paper was written by R.L., H.H., and S.U., with contributions from all co-authors. R.L. led the overall execution of the project, while H.H. provided overall supervision.

## Competing interests
The authors declare no competing interests.
