## [Peer Review File · Nature Communications]

Non-Thermal Emission in Gap-Mode Plasmon PhotoluminescenceReviewer #1 (Remarks to the Author):

In this interesting work Lemasters and co-workers investigate experimentally non linear photoluminescence emission from plasmonic structures supporting gap-plasmon modes under ultra-fast laser excitation in the near-infrared range (785 nm laser line). In particular the authors show that the non-thermal part of the hot carrier distribution plays a major role in the nonlinear photoluminescence emission. I find the system design (use of MDM structures) very clever to enable the study of this phenomenon in a controlled manner. Also, I think the topic of PL in metallic and plasmonic nanostructures is very important for the materials and nanophotonic communities. Thus, I think the paper would be suitable for Nature Communications. At the same time, however, I do have a few major concerns regarding novelty and methodology that I would like the author to address before I can recommend the paper for publication.

Major Points

- Novelty of the model. The authors strongly emphasize the fact that the non-thermal contribution from PL has been largely neglected. This is also reflected in Figure 4 in Supplementary where in column 4 they show that all prior works did not consider the non-thermal part of the hot carrier distribution. However, this contribution has been already shown for the CW PL emission (for example this paper: ACS Nano 2021, 15, 8724–8732). Moreover, Ref 22, which was very recently published, definitely accounts for the non-thermal part of the distribution when calculating the nonlinear PL. Therefore, I think the authors should provide a fairer description of their novelty. Indeed, in my opinion, this is not strictly on the non-thermal part, which should be discussed also in relation to the CW results and Ref 22, but mostly on the more controlled system and the investigation of the role of nanoconfinement VS electric field enhancement.
- Time dynamics of PL. The measurements performed by the authors are focused on the strong non-linear PL observed under pulsed excitation (150fs, 80MHz Ti:Sapp pulsed laser). Therefore, I would expect the PL signal to have a strong time-dependent evolution, with the strongest non-thermal emission at the earliest stages, changing into a thermal-emission and finally dropping to zero. However, in the measurement discussion part I could not find any reference regarding the time-dynamics of the PL signal. This makes me assume that the authors performed a continuous integration of the PL signal over time, which is understandable considering possible experimental constraints (low signal). However, as the model they use to interpret their data does not include any time-dynamics either, I am wondering whether the ratios obtained by the authors are meaningful for their experiments. In particular, the author show values of alpha in Figure 3f and 4c,f. However these are basically estimates based on a steady state PL signal with two fixed contributions, which is not representative of the real signal they have (which is the time integral of the time-varying PL signal). I would like to ask the authors to clarify this point and add adequate discussion of these aspects.

Minor Points

- In Figure 4d the authors present the results as a function of the pulse fluence. However it is not entirely clear to me how this result is distinct from the power dependent measurements from which they derived the other trends. In fact I assume that to change the power they do change the pulse fluence. Probably I am just confused but it would be helpful if the authors were clarifying this point.
- In Figure 3a we see that the transmittance spectra have a ~10% change in magnitude between different structures. Furthermore in Figure S1 it looks like the transmittance measurements are performed with back illumination while PL measurements are performed based on front illumination. While the transmittance will be the same the reflectance will likely change in magnitude between the different structures. Overall, do the authors know whether the absorptance is comparable for all the studied structures or are there significant variations that could change the PL magnitude between different structures? The PL signal should also depend on the number of absorbed photons.
- Figure 3f looks a bit strange as most dots are close to zero. I wonder whether it would be better to scale the axis so their actual value can be seen better.
- Figure S3c I think the legend should be in microW not microJ or otherwise the legend should be corrected.

Reviewer #2 (Remarks to the Author):

The report discusses how to interpret the inelastic light signal from plasmonic metals. This is a hot topic, but I think the authors are a bit behind on the state-of-the-art developments in this research area. I am also confused by several claims in the manuscript, and other important lines of evidence are missing or not discussed. I do not think the paper has much value being published in its current form. I have only outlined the major points of confusion below.

I would point the authors to recent reports from Link (<https://doi.org/10.1021/acs.nanolett.3c00622>) , Sheldon (<https://doi.org/10.1063/5.0032763>, <https://doi.org/10.1146/annurev-physchem-062422-014911>), and Tagliabue (<https://arxiv.org/abs/2307.08477>) that have advanced research in this area recently. In particular, Sheldon et al. have described the non-thermal and thermal contributions to the signal, and Tagliabue has described how the spectral ranges of the signal (in the case of gold) determines whether PL or Raman-like mechanism underlie the collected light signal. Therefore, this manuscript is not the "first" to make these claims, as emphasized in the conclusion and elsewhere.

The largest issue I have is that the analytical model put forth by these authors is never actually plotted or compared directly to the data. We only see the spectral-dependent power-law. This is a serious problem, because the reader can not assess the adequacy of the analytical model to describe the primary experimental observable. The direct comparison between the analytical model and the spectra (such as in Fig. 2d) needs to be provided.

Several studies show that the signal contains two separate thermal temperature distributions in the signal. I am wondering why or how this is omitted in the analysis, and if the signal is observed in the primary data.

I am confused about the discussion attempting to distinguish "field enhancement" versus "field confinement, since field confinement in plasmonic architectures is what provides field enhancement. Do the authors intend to express that the electron mean-free path is being modified, as it relates to scattering that relaxes the momentum-matching requirements for inter-band transitions? If so, they probably want a term like "electronic confinement".

pg. 4 - not clear how Raman-like versus inelastic PL can be physically discriminated without time-resolved studies, like in the study by Link et al. mentioned above.

Reviewer #3 (Remarks to the Author):

I have carefully read and considered the submitted paper, in which the authors devise a plasmonic sample to single out the non-thermal contribution to Au photoluminescence and demonstrate it unambiguously, also with the support of a solid modelling. The paper is timely and represents a step ahead in the understanding of electron dynamics and luminescence in metal nanostructures. The approach is rigorous and I have only minor issues to raise, which will be discussed later on in this report. For all these reasons, I am in principle in favor of publication.

However, despite the relevance of the paper, I found that its impact and applicability are not properly addressed throughout the text, especially in the abstract and in the introduction, and I believe that the authors need to reconsider the overall presentation before the manuscript can be published free of ambiguities.

In particular, when looking at the vast literature on the topic, the authors recognize that very different results have been obtained over the last two decades. However, I cannot agree with them when they refer to the 'contradictions' of such results. There is no contradiction, since they refer most often to

very different systems. From this point of view, also the submitted manuscript should in my opinion avoid any misleading (even if implicit) claim of generality and rather stress that the goal and success of this investigation was to devise a specific sample in which one of the possible contributions to the photoluminescence is enhanced and therefore properly singled out of the many available channels. This is crucial to me, since when I read the abstract I get the ambiguous message that a general explanation to Au photoluminescence will be attempted, while the manuscript is about highlighting the emergence of one specific and previously elusive contribution. So, referring to the abstract, I would not use sentences like 'to show that the nonlinear behavior is dominated by large deviations of excited carrier..', rather I would state that by a proper choice of the sample the authors were able to prove that non-thermal contributions can, under special circumstances, become dominant.

Along the same line of generality, in the first page of the introduction the authors seem to draw a picture in which only very early studies associated sub-bandgap excitation with the possibility of interband recombination, somehow suggesting that the intraband recombination is the commonly accepted relaxation channel in this case. However, other following works have also supported the same picture [see e.g. J. Phys. Chem. B 109, 13214 (2005), Phys. Rev. B 80, 045411 (2009), Nano Lett. 12, 2941 (2012)]. Moreover, the last of such work also provided evidence of a 4-photon regime which seem to fall outside the asymptotic behaviors of the model proposed in the submitted manuscript but is e.g. in agreement with the power dependence of one of the first multiphoton emissions from antennas as reported in Science 308, 1607 (2005). I believe that the authors should therefore expand the picture drawn in the introduction to further clarify that they are addressing a very specific case.

Besides this main comment, I have a few minor issues after reading the manuscript:

1) I recommend that all the spectra are represented with the same units in the horizontal axis, choosing either wavelength or energy for all of them, in order to make the comparison easier.

2) To the best of my understanding, I have difficulties in catching the argument provided in Section C. In particular, I cannot find a quick and straightforward reason why the relative thermal/nonthermal contribution α should in this case increase with the detuning, given the fact that the field confinement and therefore the weight of high-k contributions should not vary across the different measurements. This is certainly a lack of understanding from my side, but I suggest that the authors are more explicit in discussing the relevance and the rationale of this specific result.

3) Also, as a more technical note, it is not immediately clear to me why in Figs. d-f the authors provide intervals rather than specific values for the pulse energy (which, by the way, is probably mistakenly referred to as 'power fluence' in the caption).

Once all the above issues have been addressed and resolved, the manuscript can in my opinion be considered for publication. I will be happy to read it again if needed.

Point by point response to Reviewer comments.

The Reviewers' comments are in *italics*, and all changes in the revised manuscript are highlighted in red.

Reviewer #1:

In this interesting work Lemasters and co-workers investigate experimentally non linear photoluminescence emission from plasmonic structures supporting gap-plasmon modes under ultra-fast laser excitation in the near-infrared range (785 nm laser line). In particular the authors show that the non-thermal part of the hot carrier distribution plays a major role in the nonlinear photoluminescence emission. I find the system design (use of MDM structures) very clever to enable the study of this phenomenon in a controlled manner. Also, I think the topic of PL in metallic and plasmonic nanostructures is very important for the materials and nanophotonic communities. Thus, I think the paper would be suitable for Nature Communications. At the same time, however, I do have a few major concerns regarding novelty and methodology that I would like the author to address before I can recommend the paper for publication.

Response: We thank the Reviewer for the positive overall evaluation of our work and for the constructive suggestions.

1. *Major Points*

• Novelty of the model. The authors strongly emphasize the fact that the non-thermal contribution from PL has been largely neglected. This is also reflected in Figure 4 in Supplementary where in column 4 they show that all prior works did not consider the non-thermal part of the hot carrier distribution. However, this contribution has been already shown for the CW PL emission (for example this paper: ACS Nano 2021, 15, 8724–8732). Moreover, Ref 22, which was very recently published, definitely accounts for the non-thermal part of the distribution when calculating the nonlinear PL. Therefore, I think the authors should provide a fairer description of their novelty. Indeed, in my opinion, this is not strictly on the non-thermal part, which should be discussed also in relation to the CW results and Ref 22, but mostly on the more controlled system and the investigation of the role of nanoconfinement VS electric field enhancement.

Response: We agree with the Reviewer that the main novelty of our work is in the ability to control the nonthermal contribution and to achieve the experimental conditions for this contribution to become dominant. We have revised the wording to avoid the unintended and

misleading impression that we claim to be the first to consider non-thermal contributions to electron distribution. The revised abstract and the introduction highlight the broader scope of relevant PL studies including those demonstrating non-thermal carrier distribution pointed out by the Reviewer.

Changes to the manuscript:

Expanded and reworded the abstract. The revised relevant part reads,

“We use plasmonic gap-mode resonators with precise nanometer-scale confinement to show that the nonlinear PL behavior can become dominated by non-thermal contributions produced by the excited carrier population that strongly deviates from the Fermi-Dirac distribution due to the confinement-induced large-momentum free carrier absorption beyond the dipole approximation.”

Expanded the introduction suggested by the Reviewer. The revised relevant sentences now read:

Precisely controlled field confinement in the studied nanogaps enables the formation of a non-Fermi transient electron distribution driven by the plasmonic field, due to the breakdown of the dipole approximation. Our approach enables efficient control of the relative strength of thermal and non-thermal carrier contributions to PL by engineering large-momentum electronic transitions in nanostructures.

Expanded and more thoroughly acknowledged previous studies of non-Fermi contributions to emission in the section “Non-Thermal (Non-Fermi) PLE”,

...although several studies have investigated the spectral character of emission with a non-Fermi term [22, 30].

2. Time dynamics of PL. The measurements performed by the authors are focused on the strong non-linear PL observed under pulsed excitation (150fs, 80MHz Ti:Sapp pulsed laser). Therefore, I would expect the PL signal to have a strong time-dependent evolution, with the strongest non-thermal emission at the earliest stages, changing into a thermal-emission and finally dropping to zero. However, in the measurement discussion part I could not find any reference regarding the time-dynamics of the PL signal. This makes me assume that the authors performed a continuous integration of the PL signal over time, which is understandable considering possible experimental constraints (low signal). However, as the model they use to interpret their data does not include any time-dynamics either, I am wondering whether the ratios obtained by the authors are meaningful for their experiments. In particular, the author show values of alpha in Figure 3f and 4c,f. However these are basically estimates based on a steady state PL signal with two fixed contributions, which is not representative of the real signal they have (which is the time integral of the time-varying PL signal). I would like to ask the authors to clarify this point and add adequate discussion of these aspects.

Response: We thank the Reviewer for bringing up an important issue of temporal evolution. The initial non-thermal distribution is expected to thermalize on the timescale of 10-15 fs, much faster than the pulse length of 150 fs. Thus, the non-Fermi distribution towards the end of the pulse is determined by the interplay between thermalization and non-thermal optical excitation and can be quite different from the distribution at the beginning of the pulse. Unfortunately, resolving these processes temporarily is practically impossible with the existing technology, as also pointed out by the Reviewer, and is thus beyond the scope of this study.

Despite these limitations, we believe that the values of the parameter alpha characterizing the distribution extracted using our model provide a meaningful estimate for the relative contributions of thermal vs non-thermal processes. To verify this in the context of time-dependent dynamics, we have performed additional modelling of thermal emission using an exponentially decreasing carrier temperature. As is shown in the figure below, time dependence results in a slight modification of the power exponent spectrum (blue solid curve) compared to the steady temperature (red line), which can be approximated with a slightly lower effective temperature (blue dashed line). Most importantly, the curve is still linear, and the step-like behavior characteristic of the non-thermal distribution is absent. Thus, while we agree with the Reviewer that measurements and analysis of time dependence would be interesting, our data and modelling still provide an unambiguous proof for the dominance of nonthermal emission in small-gap samples, which is the main result of our work.

We have added “Dynamic Temperature Modeling” section in the SI which includes this time-dependent thermal analysis where we simulate emission from a time-varying hot-electron distribution and show that it still cannot reproduce the non-Fermi PLE we have observed.

Changes to the manuscript:

Added “Dynamic Temperature Modeling” section to SI.

3. Minor Points

• In Figure 4d the authors present the results as a function of the pulse fluence. However it is not entirely clear to me how this result is distinct from the power dependent measurements from which they derived the other trends. In fact I assume that to change the power they do change the pulse fluence. Probably I am just confused but it would be helpful if the authors were clarifying this point.

Response: The Reviewer is correct that the experimental approach used to produce Figs. 4d-f is the same as in the earlier plots for the gap and detuning dependence. In these figures, a single small excitation power range was used for each gap size/detuning value. In contrast, in Figs. 4d-f we break down the power tuning range to several smaller non-overlapping excitation ranges. This allows us to extract the dependence of the nonthermal population amplitude and effective temperature on the excitation power.

For example, analysis of the spectra for the power ranges 1-5 uW vs 101-105 uW would yield a much larger carrier temperature in the latter case. Nevertheless, Figs. 4d-f show that α , the internal state parameters, and the excitation power for a particular gap geometry are sufficient to accurately predict the resulting PLE. The shown power dependence confirms this and demonstrates the robustness of our theoretical approach.

We note that high fluence can result in photothermal melting and spectral shifting of the nanogaps due to the strong field enhancement, which may lead to measurement artifacts. Therefore, we chose lower fluence ranges for our analysis.

Changes to manuscript:

Added a sentence to Section D: Fluence-dependent PL:

To test these scaling relationships, we have repeated the PLE measurements for several distinct characteristic fluence values by breaking down the full excitation power range into narrow non-overlapping segments.

4. • In Figure 3a we see that the transmittance spectra have a ~10% change in magnitude between different structures. Furthermore in Figure S1 it looks like the transmittance measurements are performed with back illumination while PL measurements are performed based on front illumination. While the transmittance will be the same the reflectance will likely change in magnitude between the different structures. Overall, do the authors know whether the absorptance is comparable for all the studied structures or are there significant variations that could change the PL magnitude between different structures? The PL signal should also depend on the number of absorbed photons.

Response: We thank the Reviewer for this point. The Reviewer’s interpretation of our experimental setup is correct. Unfortunately, it is very challenging to probe the absorbance of our samples precisely, due to several reasons. First, the transmittance is very sensitive to the measurement location on the sample. When designing the samples, we prioritized the ease of tuning the spectral resonances, and thus used the trapezoidal nanowire geometry with a varying width along the wire allowing us to tune the linear resonances with respect to the fixed laser excitation wavelength. Consequently, the signal and the reference for the transmission must be measured at very different locations on the sample, adding significant uncertainty to the transmittance amplitude due to the minute local inhomogeneities of the substrate and the gold film, physical movement of the sample over many tens of microns etc. Second, as the reviewer notes, we also would need the reflectance spectrum for determining the absorbance. This necessitates white light (broadband) illumination through the objective. Such measurements are possible (and are performed in our lab) but are again associated with relatively significant uncertainties because of issues associated with wide field illumination and chromatic aberrations. Thus, we find that given the rapid spatial variation of the signal we can only reliably measure the spectral shapes of the resonances, whereas the uncertainties of the amplitudes are at least several percent. Therefore, our variable-width nanowire sample design, while enabling precise measurements of the nonlinear photoluminescence, is not optimal for the precise determination of absorption.

5. • Figure 3f looks a bit strange as most dots are close to zero. I wonder whether it would be better to scale the axis so their actual value can be seen better.

Response: We thank the Reviewer for this suggestion. We have changed the scaling from linear to logarithmic for all alpha plots to better illustrate these data points. For the specific values, we refer the Reviewer to Table S1 in the SI for all the parameters used in the fitting.

Changes to manuscript:

Changed all plots of alpha in Figs 2 and 3 from linear to logarithmic scaling.

6. • Figure S3c I think the legend should be in microW not microJ or otherwise the legend should be corrected.

Response: We thank the Reviewer for pointing out this oversight. We have made the correction.

Changes to the manuscript:

Changed legend in Fig S3c from “uJ” to “uW.”

Reviewer #2 :

1. The report discusses how to interpret the inelastic light signal from plasmonic metals. This is a hot topic, but I think the authors are a bit behind on the state-of-the-art developments in this research area. I am also confused by several claims in the manuscript, and other important lines of evidence are missing or not discussed. I do not think the paper has much value being published in its current form. I have only outlined the major points of confusion below. I would point the authors to recent reports from Link (<https://doi.org/10.1021/acs.nanolett.3c00622>) , Sheldon (<https://doi.org/10.1063/5.0032763>, <https://doi.org/10.1146/annurev-physchem-062422-014911>), and Tagliabue (<https://arxiv.org/abs/2307.08477>) that have advanced research in this area recently. In particular, Sheldon et al. have described the non-thermal and thermal contributions to the signal, and Tagliabue has described how the spectral ranges of the signal (in the case of gold) determines whether PL or Raman-like mechanism underlie the collected light signal. Therefore, this manuscript is not the “first” to make these claims, as emphasized in the conclusion and elsewhere.

Response: We thank the Reviewer for this comment. We agree that we should have been more careful in referencing the relevant published work and in describing the novelty of our results, which lies in the ability to control the nonthermal contribution and achieve experimental conditions when such contribution can be dominant.

We thank the Reviewer for bringing to our attention the important works that were overlooked in our original submission. We agree with the Reviewer that these papers provide a substantial contribution to the field of non-equilibrium carrier dynamics. However, they do not significantly overlap with the results of our work, which demonstrates that electron distribution is not only non-equilibrium, but also cannot be described, even in principle, by an effective temperature.

The paper by Sheldon et al refers to non-thermal carrier distributions, but these distributions are fundamentally thermal, albeit very far from true equilibrium, whose electron-hole overlap is represented by the Lorentz lineshape. Although they report much greater temperature values than the ‘hot’ carrier distributions, the carriers are still represented by a Fermi distribution. As a result, the “non-thermal” occupation is near the Fermi level which is not as useful for applications where highly energetic carriers are desired. This is not surprising since the samples used in this work are regular gold nanodisks that do not provide a mechanism to enhance the Landau damping and generate large non-thermal occupancies away from the Fermi level. In our work, we use the cryo-deposition technique to fabricate carefully controlled ultrasmooth nanogaps, which allows us to enhance Landau damping by creating “high momentum photons”, resulting in carrier distributions that cannot be approximated by the thermal Fermi-Dirac distribution.

Similarly, the preprint by Tagliabue et al. investigates phenomena that occur in the thermalized regime described by the Fermi distribution and an effective temperature. Link et al do not address the nature of the distribution of the excited carriers.

We have revised our manuscript to better acknowledge the works which have studied the effects of non-Fermi carrier distribution on emission in nanoscale structures. However, to the best of our knowledge, we are the first to explicitly show that this contribution can be controlled through the confinement of electromagnetic fields. This leads to the surprising observation of not only distinct signatures of non-Fermi carriers, but their complete dominance of emission in our narrow-gap samples. Using our nanometric MDM structures, we establish that this emission is correlated with the spatial confinement of the plasmonic gap modes.

Changes to the manuscript:

Updated and expanded introduction and added references suggested by the Reviewers,

Furthermore, these studies pointed to a plethora of possible processes dependent on the specific experimental conditions such as nanoparticle geometry and the choice of excitation source [3, 23]. Additional channels such as multi-photon absorption, hybrid interband-assisted transitions [2, 20, 24, 25] and Raman scattering [26–29] were invoked to explain the observed spectral characteristics of PL emission and their power scaling, suggesting that the underlying mechanisms may not be universal. Broadband PL was recently observed even under CW excitation, putting into question the roles of field gradients and of the overall field enhancement [19, 20, 30].

A common theme among these experiments is the requirement for large momentum transfer in intraband transitions facilitated by the field confinement.

2. The largest issue I have is that the analytical model put forth by these authors is never actually plotted or compared directly to the data. We only see the spectral-dependent power-law. This is a serious problem, because the reader can not assess the adequacy of the analytical model to describe the primary experimental observable. The direct comparison between the analytical model and the spectra (such as in Fig. 2d) needs to be provided.

Response:

We have followed the Reviewer’s suggestion and compared the PL spectra with the prediction of our model. The results for the 2 nm gap sample are shown in the figure below. The parameters for the theoretical emission spectra $\phi(\epsilon)$ are determined from the model described in the main text. The theoretical curves are corrected for the detector efficiency and normalized. No additional fitting parameters were used. As is evident from the figure below, the data is in

excellent agreement with the theoretical curves. This analysis is added as a corresponding section in the SI.

We note that our analysis in the main text focuses on the nonlinear characteristics because the spectra can be strongly influenced by artifacts. By their resonant nature, plasmonic fields in nanometric geometries are very sensitive to small imperfections. Nanoscopic material imperfections or environmental differences can lead to significantly different linear optical properties, complicating the determination of emission mechanisms. The advantage of the nonlinear regime is its robustness to these nanoscopic inhomogeneities. It reveals the internal mechanisms and state parameters while averaging out the imperfections and variations of linear characteristics such as the photonic density of states.

Changes to the manuscript:

Added “Linear Emission Spectra Fitting” section to SI.

3. Several studies show that the signal contains two separate thermal temperature distributions in the signal. I am wondering why or how this is omitted in the analysis, and if the signal is observed in the primary data.

Response: The Reviewer is correct that excited-state dynamics in noble metal nanostructures are often characterized by two temperatures describing the electron gas and the lattice, respectively. The energy flow via the electron-phonon and phonon-phonon couplings determines the dynamics, which are characterized by two relaxation times on the ps and ns time scales,

respectively. In such models the hot electron PL typically follows thermalization of electron gas via electron-phonon coupling and is characterized by the respective decay time. A third, very high effective temperature has been considered e.g. in some of the references listed by the Reviewer in an earlier comment, to model the highly non-equilibrium initial state of the carriers immediately after dephasing.

The assumption underlying such thermal models is that the deviations of the initial distribution from the Fermi distribution are relatively small and very short lived (several fs). Thus, the contribution of non-thermal distribution to the radiative recombination is assumed to be negligible. These thermal models are not consistent with our PLE spectra as they inevitably predict linear PLE lineshapes.

In our initial submission we did not consider the time dynamics of the thermalized electrons. However, when such dynamics are incorporated into the model the PLE lineshape remains linear. In the figure below we show that including such decay rate for the thermalized electrons (blue line) results only in a slightly different effective temperature of the initial estimate (red line) of the electron gas temperature. Most importantly, the step-like behavior associated with the non-thermal distribution is absent.

We have added “Dynamic Temperature Modeling” section in the in the SI to include this time-dependent thermal analysis where simulate emission from a time-varying hot-electron distribution and show that it still is not able to reproduce the non-Fermi-like PLE we have observed.

Changes to the manuscript:

Added “Dynamic Temperature Modeling” section in the SI.

4. I am confused about the discussion attempting to distinguish “field enhancement” versus “field confinement, since field confinement in plasmonic architectures is what provides field enhancement. Do the authors intend to express that the electron mean-free path is being

modified, as it relates to scattering that relaxes the momentum-matching requirements for inter-band transitions? If so, they probably want a term like “electronic confinement”.

Response: First, we must clarify that our study probes intra-band transitions, not inter-band transitions. Regarding the essence of the question, we are grateful to the Reviewer for providing us with the opportunity to clarify this point. While "field enhancement" and "field confinement" can be related for a given geometry, they are not necessarily synonymous. Field enhancement pertains to the value of the field, whereas field confinement refers to the gradient of the field mode. The primary aim of our approach is to create strongly confined electromagnetic fields, leading to the breakdown of translational symmetry, and pronounced field gradients that enable non-vertical transitions, as depicted in Fig. 1. Typically, such transitions are forbidden due to momentum conservation, given that photon momentum is negligible compared to the change in electron momentum in the intraband absorption process. However, mode confinement in our system effectively generates "large momentum photons" by lifting this momentum restriction. Alternatively, this phenomenon can be interpreted as a breakdown of the dipole approximation due to the presence of large field gradients. While free carrier absorption within the conduction band is generally forbidden given that the initial and the final states possess the same symmetry, this selection rule is lifted for quadrupolar-type transitions with an amplitude proportional to $\sim \langle i | \hat{Q} \cdot \nabla \vec{E} | j \rangle$. In our nanogap geometry, the field gradients and the characteristic wavevector of the optical field, k , in the vertical direction are determined by the gap size, x : $k \approx \pi/x$. Thus, our model assumes that this field confinement in the gap lifts the selection rule imposed by momentum conservation, resulting in step-like non-thermal carrier distributions, $\Delta\rho_{\text{nf}}$ (Fig. 1), with highly energetic electrons in the conduction band. Using plasma physics terminology, this process can be viewed as a resonant Landau damping process, where the coherent oscillations of plasmons dephase to create electron-hole pairs.

Changes to the manuscript:

We have changed the wording of the corresponding discussion in the manuscript to make this point clearer.

Precisely controlled field confinement in the studied nanogaps enables the formation of a non-Fermi transient electron distribution driven by the plasmonic field, due to the breakdown of the dipole approximation. Our approach enables efficient control of the relative strength of thermal and non-thermal carrier contributions to PL by engineering large-momentum electronic transitions in nanostructures.

5. pg. 4 - not clear how Raman-like versus inelastic PL can be physically discriminated without time-resolved studies, like in the study by Link et al. mentioned above.

Response: We agree with the Reviewer that time-resolved experiments could hypothetically determine the underlying timescales of the transient electronic distributions and the mechanisms of the nonlinear process. However, such measurements would be prohibitively challenging. First, the metal-dielectric-metal (MDM) geometry and the small plasmonic gaps with significant field enhancement lower the excitation damage threshold. Consequently, the fluences used in the study by Link et al. would lead to photothermal restructuring and melting of our samples. Secondly, while the distinction between the Raman and PL (coherent vs. incoherent) nature of the process is intriguing, the primary focus of our study is on the ability to excite non-thermal carrier distributions, which hold considerable potential for various hot-carrier processes. Finally, such measurements are still limited by the pulse width of the lasers which can be too long for distinguishing electron thermalization time scale on the order of few fs. Therefore, while we appreciate the potential value of such an experiment, we believe it would be both beyond the reach of the existing techniques and outside the scope of our study.

Reviewer #3:

I have carefully read and considered the submitted paper, in which the authors devise a plasmonic sample to single out the non-thermal contribution to Au photoluminescence and demonstrate it unambiguously, also with the support of a solid modelling. The paper is timely and represents a step ahead in the understanding of electron dynamics and luminescence in metal nanostructures. The approach is rigorous and I have only minor issues to raise, which will be discussed later on in this report. For all these reasons, I am in principle in favor of publication.

However, despite the relevance of the paper, I found that its impact and applicability are not properly addressed throughout the text, especially in the abstract and in the introduction, and I believe that the authors need to reconsider the overall presentation before the manuscript can be published free of ambiguities.

Response: We thank the Reviewer for the overall positive assessment of our manuscript. Below, we have followed the Reviewer's advice to address its impact and applicability more appropriately.

1. In particular, when looking at the vast literature on the topic, the authors recognize that very different results have been obtained over the last two decades. However, I cannot agree with them when they refer to the 'contradictions' of such results. There is no contradiction, since they refer most often to very different systems. From this point of view, also the submitted manuscript should in my opinion avoid any misleading (even if implicit) claim of generality and rather stress that the goal and success of this investigation was to devise a specific sample in which one of the

possible contributions to the photoluminescence is enhanced and therefore properly singled out of the many available channels. This is crucial to me, since when I read the abstract I get the ambiguous message that a general explanation to Au photoluminescence will be attempted, while the manuscript is about highlighting the emergence of one specific and previously elusive contribution. So, referring to the abstract, I would not use sentences like 'to show that the nonlinear behavior is dominated by large deviations of excited carrier...', rather I would state that by a proper choice of the sample the authors were able to prove that non-thermal contributions can, under special circumstances, become dominant.

Response: We thank the Reviewer for a constructive suggestion. We agree that the principal accomplishment of our work is the demonstrated capability to control and enhance the non-thermal contribution. As noted in the introduction, such contributions have indeed been discussed previously, especially in theoretical works. We have revised the introduction and the abstract to avoid a false impression of generality, and instead highlight the demonstrated previously elusive contribution.

Changes to the manuscript:

Expanded introduction and reworded abstract.

The relevant sentences in the abstract read:

“We use plasmonic gap-mode resonators with precise nanometer-scale confinement to show that the nonlinear PL behavior can become dominated by non-thermal contributions produced by the excited carrier population that strongly deviates from the Fermi-Dirac distribution due to the confinement-induced large-momentum free carrier absorption beyond the dipole approximation.”

The relevant sentences in the introduction read:

A common theme among these experiments is the requirement for large momentum transfer in intraband transitions facilitated by the field confinement.

In particular, geometric limitations of nanoparticle systems have prevented systematic probing of the effects of field confinement on NPL.

2. Along the same line of generality, in the first page of the introduction the authors seem to draw a picture in which only very early studies associated sub-bandgap excitation with the possibility of interband recombination, somehow suggesting that the intraband recombination is the commonly accepted relaxation channel in this case. However, other following works have also supported the same picture [see e.g. J. Phys. Chem. B 109, 13214 (2005), Phys. Rev. B 80, 045411 (2009), Nano Lett. 12, 2941 (2012)]. Moreover, the last of such work also provided evidence of a 4-photon regime which seem to fall outside the asymptotic behaviors of the model proposed in the submitted manuscript but is e.g. in agreement with the power dependence of one of the first multiphoton emissions from antennas as reported in Science 308, 1607 (2005). I

believe that the authors should therefore expand the picture drawn in the introduction to further clarify that they are addressing a very specific case.

Response: We thank the Reviewer for this suggestion. We have modified the introduction to avoid misconceptions about the generality of the presented findings. In the first part of the revised introduction, we discuss the case of gold films where the basic selection rules apply, so that near-infrared photons cannot be absorbed in the absence of local spatial inhomogeneities (roughness).

We are grateful to the Reviewer for bringing to our attention several important and substantial studies of nanoparticle systems, which were overlooked in the original manuscript. We agree that including them is crucial for providing the context, by highlighting the diverse range of mechanisms and decay channels observed in different nanoscopic systems.

Changes to the manuscript:

Expanded the introduction and added references, as follows.

Furthermore, these studies pointed to a plethora of possible processes dependent on the specific experimental conditions such as nanoparticle geometry and the choice of excitation source [3, 23]. Additional channels such as multi-photon absorption, hybrid interband-assisted transitions [2, 20, 24, 25] and Raman scattering [26–29] were invoked to explain the observed spectral characteristics of PL emission and their power scaling, suggesting that the underlying mechanisms may not be universal. Broadband PL was recently observed even under CW excitation, putting into question the roles of field gradients and of the overall field enhancement [19, 20, 30].

Precisely controlled field confinement in the studied nanogaps enables the formation of a non-Fermi transient electron distribution driven by the plasmonic field, due to the breakdown of the dipole approximation. Our approach enables efficient control of the relative strength of thermal and non-thermal carrier contributions to PL by engineering large-momentum electronic transitions in nanostructures.

Besides this main comment, I have a few minor issues after reading the manuscript:

3. I recommend that all the spectra are represented with the same units in the horizontal axis, choosing either wavelength or energy for all of them, in order to make the comparison easier.

Response: We thank the Reviewer for the suggestion, which was addressed in the revised manuscript.

Changes to the manuscript:

All the wavelength axes are changed to units of eV in both the main and the SI texts.

4. To the best of my understanding, I have difficulties in catching the argument provided in Section C. In particular, I cannot find a quick and straightforward reason why the relative thermal/nonthermal contribution α should in this case increase with the detuning, given the fact that the field confinement and therefore the weight of high- k contributions should not vary across the different measurements. This is certainly a lack of understanding from my side, but I suggest that the authors are more explicit in discussing the relevance and the rationale of this specific result.

Response: We believe that the most intuitive way to interpret this observation is by keeping in mind that non-thermal contribution is excited only at resonance, when the field is strongly confined, whereas the thermal contribution does not require resonant mode excitation. The Reviewer is correct that the non-thermal contribution is via the wavevector $k \sim \pi/d$ where the gap thickness d is unchanging for the spectrally detuned experiments. However, as the excitation spectrally moves away from the resonance peak the relative contribution of non-thermal contribution decreases. Meanwhile, the overall absorption also decreases as reflected by the decreased effective temperature of the thermal part, since less energy flows into the electronic system. In the limiting case, if the excitation wavelength is far away from resonance one would expect low-efficiency PL completely dominated by the thermal contribution ($\alpha=1$).

Changes to the manuscript:

Updated detuning PL section to further elaborate and clarify the role of reduced field enhancement with detuning, as follows:

Non-thermal contribution is excited only at resonance, when the field is strongly confined, whereas the thermal contribution does not require resonant mode excitation. As resonance detuning increases, the relative contribution of non-thermal contribution decreases. As a consequence, α increases and approaches unity at large detuning.

5. Also, as a more technical note, it is not immediately clear to me why in Figs. d-f the authors provide intervals rather than specific values for the pulse energy (which, by the way, is probably mistakenly referred to as 'power fluence' in the caption).

Response: The intervals provided in Figs. 4d-f indicate the ranges of excitation power for the respective curves. To determine the power-law exponent (PLE) for each emission wavelength, a range of excitation powers and the resulting emission intensities must be measured. Tracking the changes in emission intensity as a function of excitation intensity is essential to experimentally

extract the PLE. This approach enables one to extract an average electron temperature or state variable over this range. These parameters cannot be derived from a single spectrum measured at a fixed excitation power.

Power-dependent measurements were also conducted for the PLE in Fig. 3b-d and Fig. 4a. The unique aspect of the measurements in Fig. 4d-f is that we divided the power tuning range into small, non-overlapping excitation ranges. This allows us to extract the trends for the nonthermal population amplitude and the effective temperature of thermal contribution, both of which are clearly dependent on the excitation power. This approach is distinct from what is shown in the previous figures, where a single, small excitation range was used for each gap size or detuning value.

The aim of these measurements is to demonstrate that once the PLE and the internal state parameters are obtained for a particular gap geometry, our theoretical analysis enables the use of these extracted values, along with experimental excitation powers, to accurately predict the resulting PLE for different excitation power ranges (as shown in Fig. 4d). The fluence dependence validates the robustness of our theoretical approach, since it shows that PLE can be predicted based only on the known experimental parameters.

We thank the Reviewer for pointing out the oversight in the captions. We have made the appropriate changes in the figure caption.

Changes to the manuscript:

Changed Fig 4 caption to “PLE as a function of emitted photon energy extracted from the nonlinear PL, at different laser excitation fluence, as labeled.”

To test these scaling relationships, we have repeated the PLE measurements for several distinct characteristic fluence values by breaking down the full excitation power range into narrow non-overlapping segments.

Once all the above issues have been addressed and resolved, the manuscript can in my opinion be considered for publication. I will be happy to read it again if needed.

Response: We trust that the responses provided above, along with the changes in the manuscript, have adequately addressed the insightful points raised by the Reviewer.

Reviewer #1 (Remarks to the Author):

I thank the authors for addressing all of my major comments satisfactorily. I still have a couple of follow up minor points on my previous questions but these do not preclude publication.

(1) regarding the absorption, the authors have the numerical simulations that match really well the experiments. Maybe it is possible to assess the change in absorbed power through the model.

(2) regarding the time evolution of PL, if I understand correctly the author rationale is the following: (i) the change in the thermalized signal with temperature is negligible so a single effective temperature is sufficient; (ii) the measured (i.e. time integrated) signal cannot be explained with the trend of the thermalized signal; (iii) hence the non-thermal part, which occurs only at the very early stages, dominates the emission. The point that I feel is still a bit unclear in the current discussion is the time-dynamics of the non-thermal components. Indeed, as the author mention, this will rapidly decay, well-before the pump is over. So I think it would be useful to add a sentence to address this aspect too.

Reviewer #2 (Remarks to the Author):

The referee process has significantly improved this manuscript. Based on the feedback from the other referees and the response from the authors, I now feel that I have a much better understanding of the goals of the manuscript, as well scope and impact of the report. I am satisfied that the improved text better addresses the true novelty of the report, specifically, describing a situation where non-thermal PL is the dominant light emission (or inelastic scattering) pathway after intraband plasmonic excitation. I also better understand the technical discussion in this context, and I believe that initial points of concern I had have been adequately justified. I think the manuscript is now OK for publication.

Reviewer #3 (Remarks to the Author):

After reading the comments and issues raised by all three reviewers (which happened to be in fair agreement with each other and raised several common points) and assessing the response letter and the revised manuscript prepared by the authors, I find that a great deal of additional work has been done to meet the requests by the reviewers and that this action was in my opinion successful. In the new version, the main messages are now free of ambiguities, the literature framework is better defined, and some technical details of the modeling are clearer. Overall, I find that the description of this fine piece of experimental work has improved significantly and I support its publication in the current form.

Point by point response to Reviewer comments.

The Reviewers' comments are in *italics*, and all changes in the revised manuscript are highlighted in red.

Reviewer #1 (Remarks to the Author):

I thank the authors for addressing all of my major comments satisfactorily. I still have a couple of follow up minor points on my previous questions but these do not preclude publication.

Response: We thank the Reviewer for the positive overall evaluation of our work and for the constructive suggestions.

1. *regarding the absorption, the authors have the numerical simulations that match really well the experiments. Maybe it is possible to assess the change in absorbed power through the model.*

Response: We thank the reviewer for this suggestion. We agree that this in principle would be very interesting. The fitting of the photonic local density of states (LDOS) to reproduce the linear spectra does reveal meaningful information about the absorptive properties of the mode such as the peak and lineshape/FWHM. For example, the system can be analyzed as an antenna coupled quantum emitter, where the plasmon mode is treated as the optical antenna, and the electronic intraband transition as a quantum emitter. The intraband absorption rate may then be approximated with the field enhancement factors taken from the FDTD simulations performed in this study. This treatment is limited to systems which obey standard dipole selection rules, whereas this transition is fundamentally quadrupolar. We argue that the strong field gradients in the nanogaps lifts the quadrupolar selection rules, and access to the intraband resonance can be viewed as effectively dipolar.

This analysis requires accurate calculations for the isolated absorption cross section of the quantum emitter (intraband resonance). Within our theoretical framework, the momentum restrictions of intraband transitions are treated as negligible given the strong field gradients in the gap. Therefore, the intraband absorption cross section will spectrally be equal to that of the plasmonic absorption spectrum. The fits for the photonic LDOS from linear emission spectra may be assumed to follow the absorption profile via reciprocity. Then, the photonic LDOS fitting parameters may then be applied for defining the spectral shape of the isolated intraband absorption cross section.

However, this is still not all the information needed to predict an accurate absorption rate. Although we can confidently say the spectral shape of the intraband absorption cross section will follow the LDOS fits, the overall magnitude which is more crucial for an estimate is still lacking. The physical process of plasmon energy/momentum transfer is described by Landau damping/dephasing theory. A correspondence between the calculated absorption rate to the Landau damping term could in principle be applied and give an estimate of absorption. However, since we anticipate enhancement modifications to this damping rate due to the highly confined system, the problem would still become intractable. Finally, the $\Delta\rho_{nf}^0$ and T_{eff} parameters could offer a way to track the energy flow through the system and potentially give information of the total absorbed optical power. But again, the enhancement effects cause complications and restricts simplifications, as complex temporal dynamics are reasonably assumed to take place, affecting both (non) radiative decay channels, which they themselves are coupled.

We unfortunately feel we cannot confidently offer estimates of the absorptive properties of the structures given the nature of our experiments. Future endeavors will likely necessitate more robust broadband source absorption measurements, pump-probe, and further FDTD modeling. We thank the reviewer for the enthusiasm on this topic.

2. regarding the time evolution of PL, if I understand correctly the auther rationale is the following: (i) the change in the thermalize signal with temperature is negligible so a single effective temperature is sufficient; (ii) the measured (i.e. time integrated) signal cannot be explained with the trend of the thermalized signal; (iii) hence the non-thermal part, which occurs only at the very early stages, dominates the emission. The point that I feel is still a bit unclear in the current discussion is the time-dynamics of the non-thermal components. Indeed, as the author mention, this will rapidly decay, well-before the pump is over. So I think it would be useful to add a sentence to address this aspect too.

Response: We agree with the Reviewer in their characterization of the time evolution in the first two points. The third point is something we believe we must better clarify and give context with the overall scope of this study. Our results and interpretation of our model indicate that non-Fermi contributions to the emission seem to be the dominant and defining feature. We further find that optimization of this effect correlates with decreasing both gap size and detuning. What is most curious is the fact that even with the largest gap dimensions and detuning, the non-Fermi term appears to win out in its contribution to emission, captured by the α parameter trends. This likely indicates that the mechanism which maintains the non-Fermi component may require a reexamination of the lifetimes of these carriers with respect to actual experimental conditions.

It is true that in general the non-thermal part has a very short lifetime (5-10 fs) with respect to the thermalized hot carriers (~1 ps). This is an assumption also held by a substantial portion of the

references in this manuscript. However, this picture assumes a Dirac-Delta optical pulse to start this process. However, the pulse length of the excitation source is orders of magnitude longer than the characteristic lifetimes of non-thermal carriers (both normal and spatially confined). Due to a non-instantaneous pulse, it may simply be that as the pulse propagates through the nanostructures, the non-thermal carriers are continuously excited. Then, the “lifetimes” of the non-thermal components are maintained and “recycled” for times upwards of the pulse length, allowing for more time availability of radiative recombination of these non-thermal carriers. Those non-thermal carriers which don’t radiatively recombine will relax into a thermalized distribution, and as the pulse continues to propagate through the nanostructure, this process “recycles” as the non-thermal component is constantly reestablished. Thus, the hybrid carrier distribution, $\phi \sim \phi_{nf} + \phi_{th}$, should be maintained for the duration of the pulse. For this reason, it would then make sense to think of this hybrid carrier distribution having a lifetime on the order of the pulse length, assuming the pulse has an approximately constant temporal energy distribution. After the pulse tail finally passes through, the strong and confined optical fields necessary to promote non-Fermi carriers are no longer present, the recycling stops, and the carriers become thermal and decay towards ambient conditions on the order of 1 ps.

This is indeed an interesting idea. However, we feel we cannot make a definite claim in this matter with the current results, and more work is needed to conclusively show this. Unfortunately, resolving these processes temporarily is practically impossible with the existing technology, as pointed out by the Reviewer, and is thus beyond the scope of this study. We feel we can only confidently claim that the non-thermal signal is dominant, and enhancement is correlated to the spatial confinement of the modes.

Reviewer #2 (Remarks to the Author):

The referee process has significantly improved this manuscript. Based on the feedback from the other referees and the response from the authors, I now feel that I have a much better understanding of the goals of the manuscript, as well scope and impact of the report. I am satisfied that the improved text better addresses the true novelty of the report, specifically, describing a situation where non-thermal PL is the dominant light emission (or inelastic scattering) pathway after intraband plasmonic excitation. I also better understand the technical discussion in this context, and I believe that initial points of concern I had have been adequately justified. I think the manuscript is now OK for publication.

Response: We thank the Reviewer for the positive overall evaluation of our work and for the constructive suggestions.

Reviewer #3 (Remarks to the Author):

After reading the comments and issues raised by all three reviewers (which happened to be in fair agreement with each other and raised several common points) and assessing the response letter and the revised manuscript prepared by the authors, I find that a great deal of additional work has been done to meet the requests by the reviewers and that this action was in my opinion successful. In the new version, the main messages are now free of ambiguities, the literature framework is better defined, and some technical details of the modeling are clearer. Overall, I find that the description of this fine piece of experimental work has improved significantly and I support its publication in the current form.

Response: We thank the Reviewer for the positive overall evaluation of our work and for the constructive suggestions.